# Longitudinal Ensemble Integration for sequential classification with multimodal data

## Abstract

Effectively modeling multimodal longitudinal data is a pressing need in various application areas, especially biomedicine. Despite this, few approaches exist in the literature for this problem, with most not adequately taking into account the multimodality of the data. In this study, we developed multiple configurations of a novel multimodal and longitudinal learning framework, Longitudinal Ensemble Integration (LEI), for sequential classification. We evaluated LEI's performance, and compared it against existing approaches, for the early detection of dementia, which is among the most studied multimodal sequential classification tasks. LEI outperformed these approaches due to its use of intermediate base predictions arising from the individual data modalities, which enabled their better integration over time. LEI's design also enabled the identification of features that were consistently important across time for the effective prediction of dementia-related diagnoses. Overall, our work demonstrates the potential of LEI for sequential classification from longitudinal multimodal data.

## 1 Introduction

Data that are both longitudinal/temporal and multimodal are increasingly being used in combination with machine learning for forecasting, especially in medical diagnosis (Brand et al., 2019; Zhang & Shen, 2012; Feis et al., 2019; Li et al., 2023). Recently, a number of promising approaches for sequential classification from such data have been introduced (Eslami et al., 2023; Zhang et al., 2011; Wang et al., 2016; Zhang et al., 2024). For instance, some approaches have used recurrent neural network (RNN)-based models applied to data sequences where the modalities at each time point have been concatenated into a long feature vector, sometimes referred to as early fusion (Nguyen et al., 2020; Olaimat et al., 2023; Maheux et al., 2023). Generally, due to the complexity of this modeling, these approaches generally consider only a limited set of modalities, samples, or time points. In addition to computational issues, the early fusion of data across modalities may obfuscate signals local to the individual data modalities (Zitnik et al., 2019; Li et al., 2022). This may hinder model performance, as the differing semantics and scales of the various modalities are not adequately accounted for. This represents a major challenge for the automated classification of multimodal longitudinal data, as the consensus and complementarity among different modalities may not be sufficiently leveraged (Zitnik et al., 2019; Li et al., 2022).

Recently, the Ensemble Integration (EI) framework (Figure 1) was developed to leverage modality-specific information during data integration and predictive modeling (Li et al., 2022). EI accomplishes this by first capturing information local to the individual modalities into effective base predictors of the target outcome. These base predictors are then aggregated into final heterogeneous ensemble-based predictive models using methods such as stacking (Sesmero & Ledezma, 2015). Due to its flexibility, EI has been effective for a variety of biomedical prediction problems, such as protein function prediction and prognosis of COVID-19 patients in Li et al. (2022), detection of cell division events in Bennett et al. (2024), screening of diabetes among youth in McDonough et al. (2023), and the early detection of dementia from only snapshot baseline data in Cirincione et al. (2024). Additionally, EI has provided insights into the prediction problems in the applications listed above, e.g., risk factors of diseases, through its model interpretation capability (Li et al., 2022). Despite the successes of EI, it has thus far only been applicable in its design to non-longitudinal multimodal data.

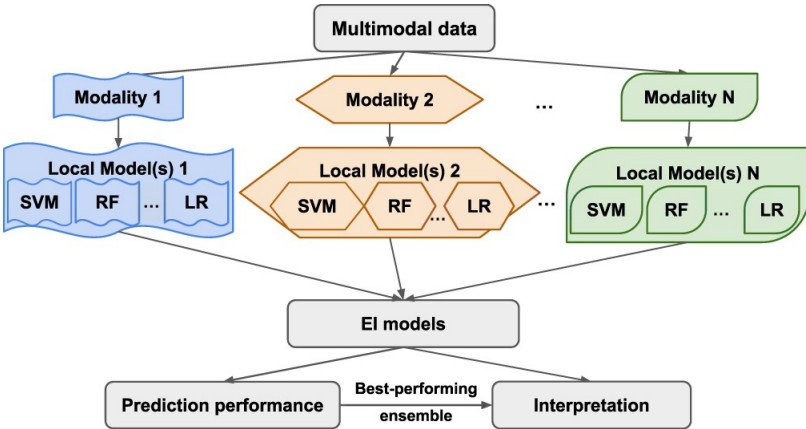

Figure 1: Overview of the Ensemble Integration (EI) framework (Li et al., 2022). In the first step, libraries of base predictor models are applied to each distinct data modality. These base predictions are then integrated using ensemble algorithms such as stacking, Sesmero & Ledezma (2015), for final classification. Interpretation of both the base predictors and the ensemble is used to determine the most predictive features.

In the current work, we extended the capabilities of EI to multimodal longitudinal data by combining its modality-specific base predictors with temporal deep learning methods like RNNs. Specifically, in the new Longitudinal Ensemble Integration (LEI) framework, we first derive base predictors from individual modalities at every time point and then stack these predictions using a Long Short-Term Memory (LSTM) Network (Hochreiter & Schmidhuber, 1997). We also leveraged EI's interpretation strategy found in Li et al. (2022) to identify the most predictive features at different time points, as well as longitudinal patterns among these features.

We tested the capabilities of LEI for the early and accurate prediction of the progression of dementia, due to this being one of the most commonly tackled problems in the sequential classification of multimodal data (most of the references cited above), as well as for its real world importance (Chen et al., 2021; Bredesen, 2014). For this, we used data from The Alzheimer's Disease Prediction of Longitudinal Evolution (TADPOLE) Challenge, Marinescu (2019), a structured dataset derived from The Alzheimer's Disease Neuroimaging Initiative (ADNI) (Petersen, 2010). TADPOLE contains multiple data modalities, e.g., demographics and cognitive test scores, as well as domain-relevant features derived from MRI images, such as volumes of brain regions, collected over the course of ADNI participants' enrolment in the study. Our specific task was to use the TADPOLE data to predict whether a patient would be diagnosed with different stages of this disorder, namely cognitively normal (CN), mildly cognitively impaired (MCI), or dementia, at the next TADPOLE/ADNI visit, given data up to and including the current one. We evaluated the performance of various configurations of LEI for this task, as well as compared its performance to those of existing approaches for multimodal sequential classification. We also leveraged EI's interpretation strategy introduced in Li et al. (2022) to identify the most predictive features for this problem at different time points, as well as temporal patterns among these features. Although demonstrated here for early dementia detection, our approach is general with respect to applications, modalities, and constituent models, and can be adapted for other data integration-based longitudinal prediction problems as well.

## 2 PROPOSED APPROACH

Below, we describe the methods used in our study. All the code used to implement these methods is available at the anonymized LEI GitHub repository.

### 2.1 LONGITUDINAL ENSEMBLE INTEGRATION (LEI)

Our proposed approach, LEI, extends the capabilities of EI (Figure 1) for sequential classification from longitudinal multimodal data. In LEI, we first apply base predictors to data from multiple modalities at the studied time points (top half of Figure 2; see Li et al. (2022) for the implementation details of EI used in this work). Longitudinal models for the target label(s) are then built upon the probabilities from the base predictors using stacking (Sesmero & Ledezma, 2015) (bottom half

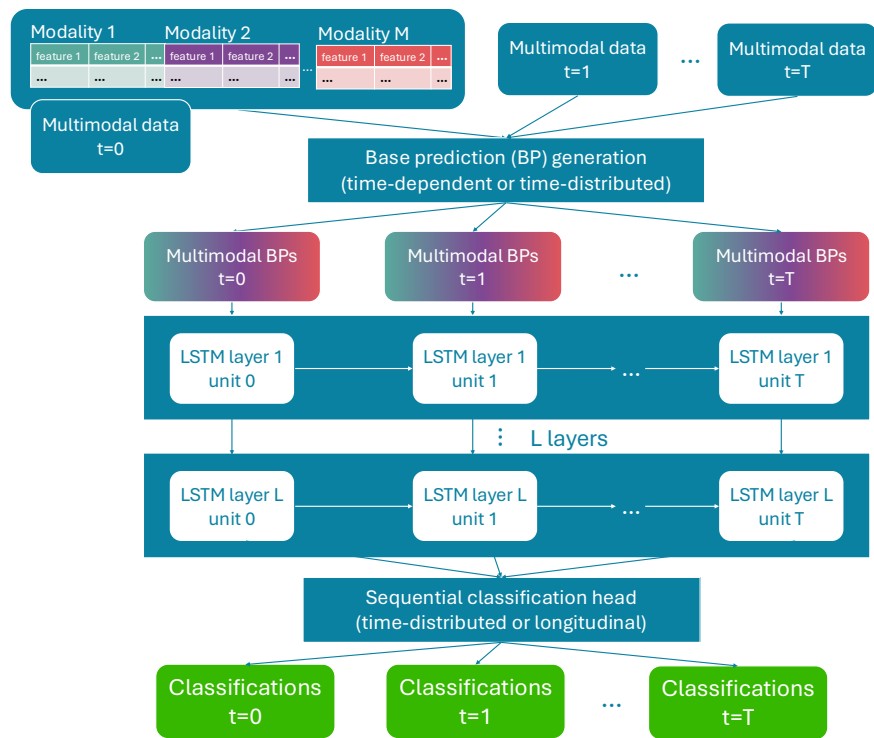

Figure 2: Overview of the proposed Longitudinal Ensemble Integration (LEI) framework. See Section 2.2 and associated figures for details of the time-dependent, time-distributed and longitudinal modeling configurations of LEI tested in this work.

of Figure 2). Stacking traditionally learns a static meta-predictor over the base predictors using any applicable classification algorithm, e.g., SVM and Random Forest, as is done in EI. However due to the longitudinal aspect of the current work, a sequence-to-sequence LSTM was used as the stacking algorithm. Specifically, in this work, each set of multiclass base predictors (such as KNN, Logistic Regression, SVM, Random Forest, and XGBoost), derived from the individual TADPOLE modalities, output probability vectors of a length equal to the number of classes, indicating whether a patient would receive a diagnosis of CN, MCI or Dementia at the corresponding time point. An ordinal representation of the labels by way of the mapping CN $\to$ 0, MCI $\to$ 1, Dementia $\to$ 2 was used for training the base predictors. In our stacking architecture, the hidden states produced by all the LSTM nodes up to and including time point $t$ were processed to produce the predicted label(s) at time point $t+1$. We took the sequence-to-sequence approach to reduce computational complexity and build models whose parameters could be optimized by complete sequences of longitudinal data.

We tested LEI with the categorical cross-entropy (CCE) loss for the constituent LSTM. Specifically, if the true label of a sample $x$ is one-hot encoded in a vector $y$ whose length is the same as the number of classes, $C$, then this loss is defined as $\text{CCE}_t(y, \hat{y}) = -\sum_{c=1}^{C} y_c \log(\hat{y}_c)$, where $\hat{y}$ is the vector of probabilities of $x$ being in each of the $C$ classes that is output by the LEI model at a given time point $t$. Standard practice in the longitudinal setting would then be to sum up all such losses over time, i.e. $\text{CCE}(y, \hat{y}) = -\sum_{t=1}^{T} \sum_{c=1}^{C} y_c^t \log(\hat{y}_c^t)$, where $T$ is the number of time points, $\hat{y}$ and $y$ are matrices representing the true and predicted labels across time respectively, and $y^t$ and $\hat{y}^t$ are row vectors representing the true and predicted labels of $x$ at time $t$, respectively.

While we could use this standard time-dependent CCE loss, it assumes that the classes being predicted are equally distributed over time, which may not be the case. To address this potential challenge we employed a class-weighted version of the time-dependent CCE loss using weights $w_c^t = \frac{N}{C \cdot n_c^t}$, where $N$ is the size of the training set and $n_c^t$ is the number of samples in the training set in class $c$ at time $t$. Another potential problem here is that the labels may be ordered, as is the case with our target problem, as each of them represents a progressing stage of neurodegeneration (CN $\to$ MCI $\to$ Dementia). It was suggested previously that an ordinal weight could be introduced,

which further penalizes a prediction arising from $\hat{y}_{\max} =$ the argmax of the predicted probability prediction vector, based on how many classes away from the true ordinal label, $y$, $\hat{y}_{\max}$ is (Hart, 2017). This weight is defined as $w_o(\hat{y}, y) = \frac{|\hat{y}_{\max} - y|}{C-1} + 1$. Using this, we arrived at a final double weighted CCE loss (DWCCE) given by

$$\text{DWCCE}(y, \hat{y}) = -\sum_{t=1}^{T} \sum_{c=1}^{C} w_o(\hat{y}^t, y^t) \cdot w_c^t \cdot y_c^t \log(\hat{y}_c^t) \tag{1}$$

This loss function for unbalanced ordinal classes is another contribution of our work that may be useful in other similar scenarios.

## 2.2 LEI CONFIGURATIONS

Although the basic design of LEI is straightforward, (Figure 2), the availability of several base predictors at multiple time points, as well as the architecture of the LSTM stacker, lend the design to be implemented in multiple configurations. Four such configurations were developed and evaluated. They are distinguished by the modeling approaches taken at the beginning base predictor step and the final classification. In the regime of sequential classification with machine learning, three broad strategies are common.

- What we refer to as *time-dependent* modeling involves setting up and solving a separate modeling problem at every time point within a sequence of data and therefore only uses a cross section of the data for model training and evaluation of all models involved.

- In the *time-distributed* modeling approach, a single static model is trained and evaluated on all of the cross sections of data from all time points within a sequence of data but separate time points are treated as independent.

- In *longitudinal modeling*, a model capable of leveraging longitudinal patterns in a sequence of data is used to classify the elements of the sequence, such as with RNNs or LSTMs.

In this work, different combinations of these three strategies were used at different steps in the LEI algorithm to yield the four combinations evaluated:

1. Time-dependent base predictors (BPs) that are stacked by an LSTM with a time-distributed classification head.

2. Time-dependent BPs that are stacked by an LSTM with a longitudinal classification head.

3. Time-distributed BPs that are stacked by an LSTM with a time-dependent classification head.

4. Time-distributed BPs that are stacked by an LSTM with a longitudinal classification head.

Below, we describe each of these configurations of LEI in detail.

### 2.2.1 CONFIGURING BASE PREDICTION GENERATION/AGGREGATION

For generating base predictions in the first step of LEI (top half of Figure 2), both time-dependent modeling (Configurations 1 and 2) and time-distributed modeling (Configurations 3 and 4) were explored. In the time-dependent approach, separate sets of base predictors were trained and used to generate predictions at each time point, followed by their concatenation for the final ensemble (Figure 3). This approach has the potential advantage of generating predictions for the stacker that are optimized for the time point corresponding to the original data.

In the time-distributed approach, the longitudinal data were flattened across the time dimension and a single instance of each of the base predictors was trained on all the data across all time points (Figure 4). A numerical feature indicating the time point corresponding to a sample was concatenated to the original data as a form of positional encoding for this approach as well. In all data splitting steps, feature vectors of the same sample were always kept in the same split to prevent data leakage. A practical advantage of this approach is that each base predictor is trained on $T \cdot N$ samples, as opposed to $N$ in the time-dependent case. Similarly, a factor of $T$ fewer total models are trained in the time-distributed base predictor approach than in the time-dependent setting, which can reduce computational complexity. A more subtle potential strength of time-distributed base predictors is

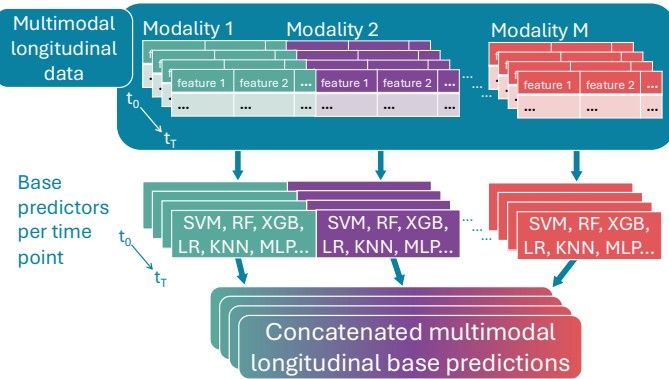

Figure 3: Time-dependent base predictor generation. Separate instances of the modality-specific base predictors are trained on data from distinct time points. The base predictions are then concatenated into an intermediate longitudinal dataset for stacking using an LSTM.

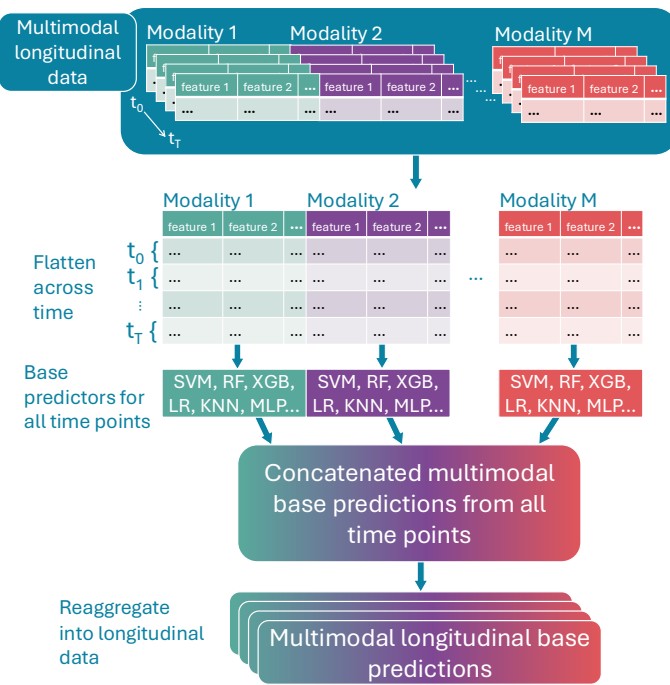

Figure 4: Time-distributed base predictor generation. Single instances of the modality-specific base predictors are trained on data from all time points. The base predictions are then concatenated into an intermediate longitudinal dataset for stacking using an LSTM.

that training the same instances of each model on all of the longitudinal data within a modality guarantees the semantic consistency of the base predictions when used as longitudinal features for classification by the downstream LSTM. This is because for a given modality/base predictor pairing, the corresponding feature at every time point represents a prediction arising from the same decision boundary. The semantic consistency of features across time is an essential aspect of the inductive bias of models like RNNs and LSTMs (Hochreiter & Schmidhuber, 1997).

### 2.2.2 CONFIGURING THE SEQUENTIAL CLASSIFICATION HEAD WITHIN LEI

While LSTMs were used in all configurations of LEI, different approaches were taken for the final sequential classification of the data. The standard approach, as in Foumani et al. (2024), employs time-distributed modeling where the hidden states output at every time point by the LSTM are processed by a classifying MLP (Configurations 1 and 3). This approach has the possible benefit

of consistently strong performance across time due the model's focus on feature level information independent of time point. There is also the general possibility of modest gains across time due to the richer temporal information stored in hidden states from later time points.

We also employed a longitudinal modeling approach at this stage, where a multi-layered LSTM was used. The softmax activated hidden states from the last layer, equal in length to the number of classes, were output to the loss function (Configurations 2 and 4). This approach may have the advantage of improved performance across time as the classifier can capture temporal dependencies and use them to make a classification.

A time-dependent sequential classification approach was also explored. However, this approach did not perform on the same level as the other approaches due to the limited training data available to the classifiers at each time point.

A final variation tested for LEI concerned the labels the base predictors were trained on, given the goal of making predictions at the next time point. Two reasonable choices would be to match the data from time point $t$ either to the corresponding label at time $t$ or to the label at time point $t+1$. We found that the $t$ to $t$ approach outperformed the $t$ to $t+1$ approach in all LEI configurations, possibly because the LSTM stacker covered the $t$ to $t+1$ dependence. Thus, in all the results presented, base predictors were trained with the labels at the same time point, i.e., the $t$ to $t$ approach.

### 2.3 Interpretation of Longitudinal EI Models

The original EI framework enabled the identification of the most predictive features at individual time points. Since deep learning methods like LSTMs are well-known to be hard to interpret, as demonstrated by Zhang et al. (2021), we adopted an alternate approach based on the interpretation of static EI models (Li et al., 2022). Specifically, we used the latter's algorithm to identify the ten most predictive features at each time point, and then analyzed how these sets varied over the temporal trajectory. To make the interpretations consistent with the LEI algorithm, the stacking algorithms used in static EI were trained with the labels at time $t + 1$.

## 3 Evaluation Methodology

### 3.1 Dataset for Evaluating LEI

The longitudinal multimodal data used to evaluate LEI were from (Marinescu, 2019)'s TADPOLE Challenge , which were derived from the ADNI 1, 2, and GO studies (Petersen, 2010; Beckett et al., 2015; Toga & Crawford, 2015). These data modalities included cognitive test scores, demographic information and neuroanatomical measurements from MRI and PET scans. However, several of the original TDAPOLE modalities included features that were missing for a substantial fraction of the patients at one or more time points, which would have been difficult to impute reliably, as shown in Dziura et al. (2013), and were likely to adversely affect downstream analyses (Nijman, 2022). So the features that were missing in at least 30% of the patients at any time point were removed from our dataset. This also resulted in some modalities, such as PET scans, being excluded from our analyses. The remaining modalities that were used in our analyses were main cognitive tests, demographics, APOE4, and other characteristics, MRI volumes, and MRI Regions of Interest (MRI ROI). The last modality dominated the full set of features (313/337) which could have artificially dominated predictive modeling. To address this issue, we split the MRI ROI modality into five sub-modalities

| Modality | Feature Count |
|---|---|
| Main Cognitive Tests | 9 |
| MRI Volumes (general brain regions) | 7 |
| Demo, APOE4, and others | 8 |
| MRI Specific Regions of Interest (ROI): Volume | 40 |
| MRI ROI: Volume (Cortical Parcellation) | 69 |
| MRI ROI: Surface Area | 68 |
| MRI ROI: Cortical Thickness Average | 68 |
| MRI ROI: Cortical Thickness STD | 68 |

Table 1: Details of the longitudinal modalities of TADPOLE data used to evaluate LEI.

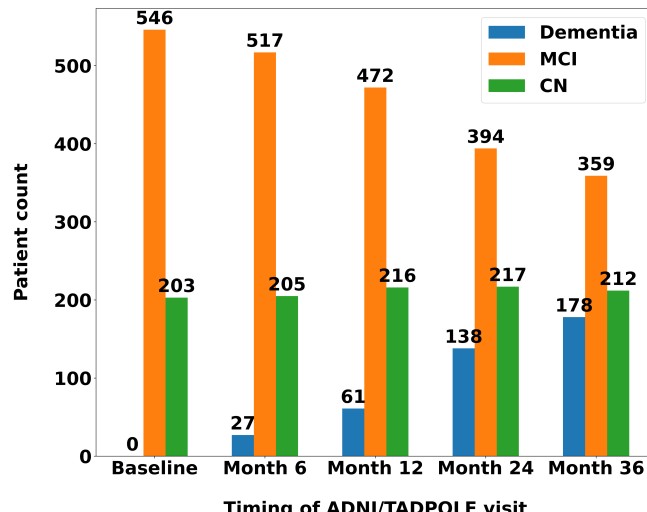

Figure 5: Distribution of patient diagnoses across ADNI/TADPOLE visits.

based on the semantics of the constituent features. Missing values of features in the resultant eight modalities (Table 1) were imputed using K-Nearest Neighbor imputation (KNNImpute) (Troyanskaya, 2001) with K = 5 within each modality at each time point. Categorical features like sex were treated as continuous ones using their original values in TADPOLE to reduce dimensionality. The one exception was the APOE4 allele, which was one-hot encoded due to its close association with Dementia risk (Kivipelto, 2008; Safieh et al., 2019).

Using this dataset, our target prediction problem for evaluating LEI was to predict how these patients' diagnoses (cognitively normal (CN), mildly cognitively impaired (MCI), or dementia) evolved over time based on data collected from them at regular intervals within the first three years of their enrollment in ADNI. Specifically, patient data at the baseline, month 6, month 12 and month 24 visits were used to predict diagnoses at the respective next visits at month 6, month 12, month 24 and month 36 respectively. Since dementia is irreversible, we only included patients diagnosed as either CN or MCI at their first (baseline) ADNI visit. The class distribution over time in our final cohort of 749 patients is shown in Figure 5.

### 3.2 TRAINING AND EVALUATION

We translated the nested cross-validation setup for the training and evaluation of the static EI framework introduced in (Li et al., 2022) to the various configurations of LEI as well. Specifically, the overall evaluation of LEI was conducted in a five-fold cross-validation (CV) setup, where 80% of the cohort was used for training the models, and the remaining 20% for evaluation. Furthermore, the base predictors at each time point were trained using the diagnosis labels at the same time point, and used to make predictions in our inner five-fold CV setup within the training split of each cross-validation round. These base predictions were then input to the LSTM stacker for training, and the predicted outputs over the five outer CV folds were concatenated for evaluation. The accuracy of LEI's predictions was measured in terms of the multi-class version of the F-measure, also known as the argmax measure (Opitz & Burst, 2021). Specifically, at each time point, we evaluated the model in terms of the arithmetic mean of the F-measure of the three classes (CN, MCI, or Dementia) after assigning the class predicted with the highest probability (argmax) as the most likely diagnosis (Berger & Guda, 2020). We also repeated the CV process 20 times to calculate the median F-measure as the representative metric, along with the standard errors indicating the statistical robustness of the performance assessments.

### 3.3 BENCHMARKS FOR ASSESSING LEI'S PERFORMANCE

Finally, to assess if LEI produced more accurate predictions of future patient diagnoses than other methods, we compared its performance to two multi-layered LSTMs, one equipped with a time-distributed sequential classifier and the other with a longitudinal sequential classifier. In both instances, the LSTM had exactly the same architecture and parameters as the corresponding stacker used in LEI. We also compared LEI to Predicting Progression of Alzheimer's Disease (PPAD)

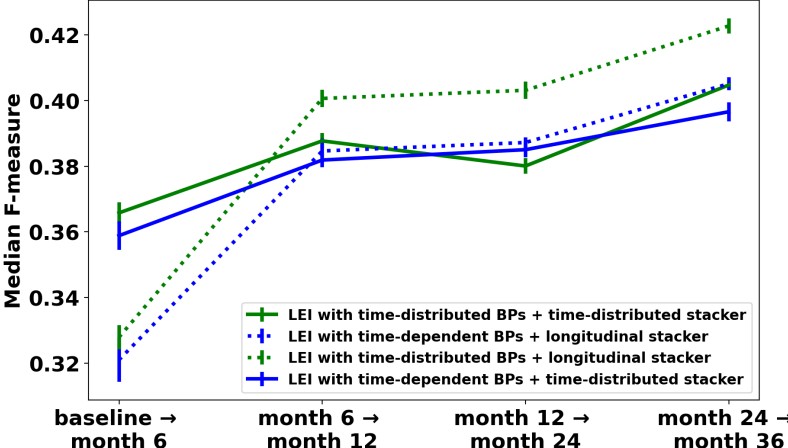

Figure 6: Performance of LEI across all base prediction generation and stacking configurations. Shown here is the performance of each configuration for predicting the label (CN, MCI or dementia diagnosis) at the next time point, using longitudinal multimodal data available until the current time point. Longitudinal sequential classifiers are shown with dotted curves and time-distributed sequential classifiers are shown with solid curves. Longitudinal classifiers are shown with dotted curves and time-distributed classifiers are shown with solid curves. BP=base predictor.

(Olaimat et al., 2023). 2023). PPAD is an RNN-based architecture originally proposed for predicting the progression of Alzheimer's Disease (AD). PPAD creates a latent representation of longitudinal clinical data, which is fed to a Multi-Layer Perceptron (MLP) to predict the diagnosis at the next visit or future visits ahead. The original PPAD model was a binary classifier to predict whether an MCI patient would convert to AD or not in a future visit. In this study, PPAD was modified into a multiclass classifier to predict CN, MCI and dementia diagnoses at each visit. We also modified PPAD to do sequential classification. Since these benchmark methods do not explicitly consider multimodal data, we concatenated all the features of the TADPOLE modalities considered for use with LEI to make the resultant data usable with these benchmark methods. The LSTMs in LEI and all the benchmarks were built in Keras (Chollet et al., 2015).

## 4 RESULTS

Below, we describe our observations on the performance and interpretation of LEI.

### 4.1 RELATIVE PERFORMANCE OF LEI CONFIGURATIONS

Figure 6 shows how the performance of LEI was affected by altering its configuration in the ways described in Section 2.2 and evaluated as specified in section 3.2. The results show how effective each configuration is at using the longitudinal multimodal data available until a particular time point to make a diagnosis prediction at the next time point. It can be seen from these results that a time-distributed approach to base predictor generation was preferable in this study, with respect to both stacking approaches, but especially when combined with a longitudinal stacker and at later time points when more longitudinal data could be leveraged (dotted green curve in Figure 6). Here, the advantage of training each base predictor on $T$ times the number of feature vectors as in the time-dependent approach, (Figures 3 and 4) may be responsible for improved downstream performance. Similarly, as explained in Section 2.2.1., time-distributed modeling guarantees semantic consistency across time in the base predictions when they are used as features in the downstream longitudinal modeling.

An observation of note is the different behaviors that arise from the use of a longitudinal classifier and a time-distributed classifier at the stacking step. Configurations with longitudinal stackers consistently performed weaker at the earlier time points but improved significantly in performance over time, eventually outperforming the time-distributed configurations. This was likely due to the longitudinal stackers' ability to aggregate information across time, whereas the time-distributed stackers

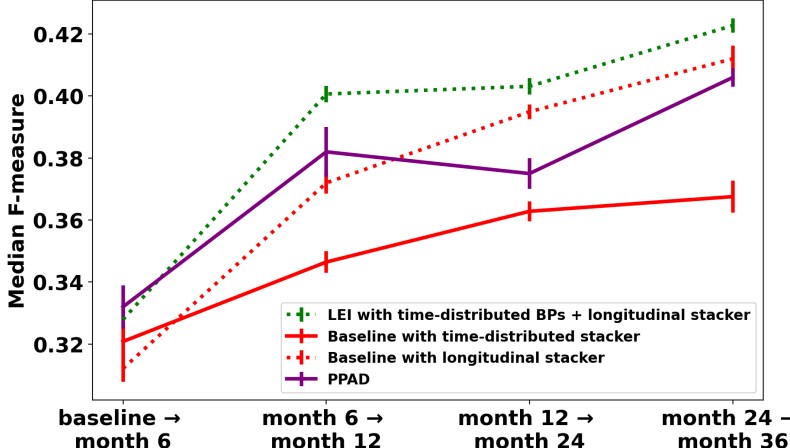

Figure 7: Performance of best LEI configuration relative to PPAD and two benchmark LSTM-based baseline models. Shown here is the performance of each method for predicting the diagnosis label at the next time point, using longitudinal multimodal data available until the current time point. BP=base predictor, LSTM=long short-term memory, PPAD=Predicting Progression of Alzheimer's Disease (Olaimat et al., 2023)

were able to maintain a consistent performance at all time points due their focus on feature level information contained in the hidden states while being agnostic to the time points the hidden state vectors corresponded to. It is also worth noting that the time-distributed models do marginally increase in performance across time, although not consistently. This may be attributable not to the architectures of the classifiers but the quality of their inputs. Specifically, hidden state vectors corresponding to later time points have richer temporal information embedded within them and so can help to make more accurate predictions.

## 4.2 COMPARISON OF LEI AND BENCHMARK METHODS

Figure 7 shows the performance of the best performing configuration of LEI (time-distributed base predictors + longitudinal stacker, see dotted green curve in Figure 6) with respect to that of the benchmarks described in Section 3.3. These results further demonstrate the ability of longitudinal sequential classifiers to aggregate information across time for increased classification performance. The baseline LSTM + MLP classifier (represented by the solid red curve), that both ignored the multimodality of the data and used time-distributed classification, performed significantly worse than all others over time. Specifically, due to its ability to leverage the complementarity and consensus of multimodal data over time in the form of modality-specific base predictors, LEI performed better than the other benchmarks, including the PPAD method proposed for a closely related problem.

## 4.3 INTERPRETATION THE LEI-BASED EARLY DEMENTIA DETECTION MODEL

Finally, we interpreted the best-performing LEI model as described in Section 2.4. The interpretation was done with respect to what was previously described as the best LEI configuration. Figure 8 shows the results of this interpretation in terms of the ten most predictive features at each time point. Our data driven algorithm showed and confirmed that CDR-SB is one of the top predictors of future cognitive outcome in line with the prior literature in Alzheimer's research (Tzeng et al., 2022; Williams et al., 2013). Also in agreement with the literature in this field, we found Entorhinal cortical thickness and volume to be major contributors to predicting patients' future diagnoses (Igarashi, 2023; Newton et al., 2024; Gómez-Isla et al., 1996; Astillero-Lopez et al., 2022; Bobinski et al., 1999). Perhaps most interestingly, the importance of the Functional Activities Questionnaire (FAQ) increased at later time points. This is consistent with the importance of FAQ as a key examination in differentiating MCI from dementia when the cognitive evaluations are similar (Marshall et al., 2015; Teng et al., 2010). Thus, it is sensible that the importance of this feature increased as LEI was trying to make more predictions of dementia cases at later time points (Figure 5). Observations such as the above suggest the utility of LEI for uncovering useful domain knowledge about longitudinal multimodal prediction problems like the early detection of dementia.

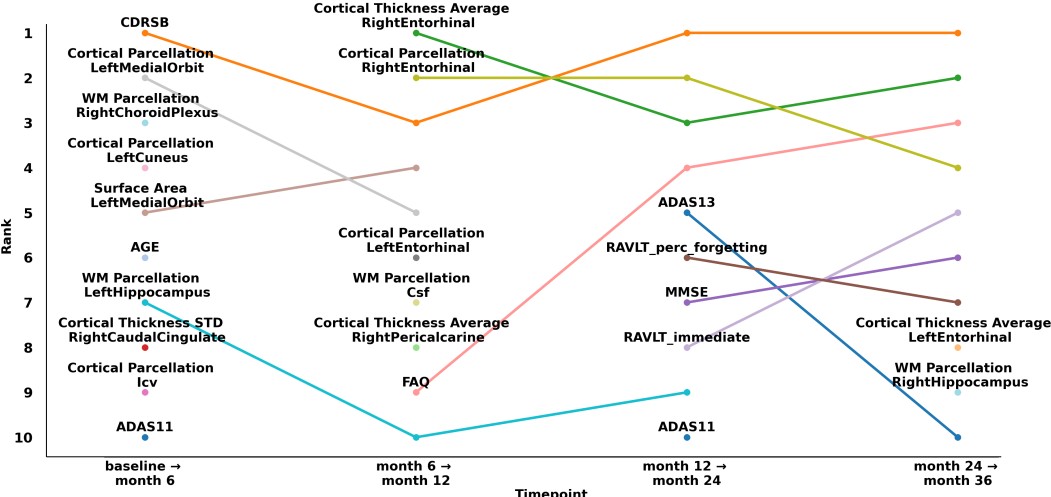

Figure 8: Top 10 most important features at every time point for making diagnosis predictions at the subsequent time point using LEI. If a feature appears in the top 10 at consecutive time points, we draw a curve between the relevant points in the plot.

## 5 DISCUSSION

This work investigated the potential of our novel Longitudinal Ensemble Integration (LEI) framework, for sequential classification from temporal multimodal data. LEI builds upon the success of the existing EI framework by integrating base predictors inferred from the multimodal data over time through an LSTM stacker. We tested LEI on longitudinal multimodal clinical data from the ADNI-derived TADPOLE cohort to predict the likelihood of patients' progression to dementia. LEI performed better than several other approaches that have been used for predicting Dementia progression, such as an LSTM and PPAD applied to the raw TADPOLE data. The framework also identified several predictive features and their variations over time that could expand our knowledge of the key characteristics of progression to dementia. In conclusion, our work demonstrates the potential of LEI for effectively integrating longitudinal multimodal data for sequential classification.

However, our work also had some limitations. During our processing of the TADPOLE data, any feature with missing values for over 30% of patients at any time point was eliminated. This resulted in several clinically useful imaging modalities, such as FDG PET, AV45 PET, AV1451 PET and DTI, being removed, potentially deteriorating prediction performance. The exclusion of the non-longitudinal features within TADPOLE data may also have caused a reduction in predictive performance. Some limitations related to the label set (CN, MCI and Dementia) were notable as well. The considerable imbalance in the class labels and its variation across time (Figure 5) could have biased our predictive models in favor of the more prevalent classes/diagnoses. Although we attempted to address this challenge by designing a weighted ordinal cost function for training the LSTM within LEI, the problem still affected performance.

Additionally, although the general design of LEI makes it capable of being used for both structured and unstructured data, our work only considered structured modalities in the TADPOLE data. Future work can include the development and validation of LEI for collections of structured and unstructured data, such as those available from ADNI. Future work can also include the consideration of variable length sequences of longitudinal data, which are typical in large cohorts like ADNI, and which LSTMs and other longitudinal deep learning models are capable of processing. Other potential configurations of the LEI framework can be explored as well. The use of longitudinal modeling at the base prediction step could further enhance the benefits of LEI by better leveraging the temporal information within modalities. Similarly, in both the time-dependent and time-distributed base predictor formulations, training base predictors on feature vectors that are not necessarily part of an unbroken sequence of data may be useful in strengthening base predictions, as these modeling approaches do not require sequential data, even if these samples can not be used in the ultimate sequential classification task.

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
