# OpenReview forum: "Longitudinal Ensemble Integration for sequential classification with multimodal data"
_ICLR.cc/2025/Conference — Submitted to ICLR 2025_

### Official Review · Reviewer_Bm11 · 2024-10-25

**Soundness:** 2
**Presentation:** 2
**Contribution:** 2
**Rating:** 3
**Confidence:** 4

**Summary:**

This research aims to address the challenge of effectively modeling multimodal longitudinal data. This paper introduces a novel framework called Longitudinal Ensemble Integration (LEI), which extends the existing Ensemble Integration (EI) framework by incorporating RNN networks. The results indicate that LEI outperformed benchmark models in predicting dementia progression.

**Strengths:**

(1) The proposed LEI framework outperforms existing methods in disease prediction based on multimodal longitudinal data.
(2)  The proposed LEI facilitates the identification of consistently important features across time.

**Weaknesses:**

(1) Novelty: The methodological contribution is limited as the LEI is the combination of EI and LSTM. There are many other temporal deep learning methods (e.g. Transformer-based models), which prove to be better than LSTM. However, this paper does not discuss their compatibility for EI.
(2) Clarity & Explanation: The writing contains many unclear phrase and what seem to be translation. E.g., "Four such configurations were the same sample were always kept in the same split to prevent data leakageeveloped and evaluated..." , "sometimes referred to as early fusion"

**Questions:**

See weaknesses.
My primary concern is that this paper lacks novelty and low writing quality makes it hard to read.

---

### Official Review · Reviewer_AHTd · 2024-10-31

**Soundness:** 2
**Presentation:** 2
**Contribution:** 2
**Rating:** 3
**Confidence:** 4

**Summary:**

This paper introduces a Longitudinal Ensemble Integration (LEI) framework for sequential classification using multimodal longitudinal data.  LEI integrates modality-specific base predictors with an LSTM stacker to capture temporal patterns and improve prediction accuracy across time. Experiments on the Alzheimer’s Disease Prediction of Longitudinal Evolution (TADPOLE) dataset demonstrate that LEI outperforms an existing Predicting Progression of Alzheimer’s Disease (PPAD) approach by leveraging weighted loss functions and multiple configurations to handle data imbalance and optimize multimodal integration.

**Strengths:**

1. LEI introduces a novel double-weighted loss function to handle class imbalance and ordinal labels.
2. The framework tests four configurations, combining time-dependent and time-distributed base predictors with various classification heads. This allows for optimized multimodal integration and improved handling of complex longitudinal patterns.

**Weaknesses:**

1. LEI builds on the previous EI framework but follows an intuitive approach to generating and aggregating base predictors, resulting in limited novelty in its integration method.
2. The evaluation of LEI is constrained, relying solely on the TADPOLE dataset and one primary comparison method (PPAD), which limits the framework's generalizability across diverse datasets.
3. The F-scores achieved by LEI range from 0.32 to 0.42, which are relatively low and may suggest that the problem is not yet well addressed.
4. The reliance on F-score alone for evaluating performance is insufficient, as it may not fully capture the framework’s effectiveness across different aspects of the problem.
5. According to the paper, the data imputation step occurred before the training/testing split, which risks data leakage by allowing information from the test set to influence training data. This can lead to inflated performance metrics and reduce the model's real-world validity.

**Questions:**

See weaknesses.

---

### Official Review · Reviewer_eC6n · 2024-11-04

**Soundness:** 2
**Presentation:** 3
**Contribution:** 2
**Rating:** 3
**Confidence:** 3

**Summary:**

The paper introduces a novel conditional generative model designed to simulate longitudinal data sequences for neurodegenerative diseases like Alzheimer’s disease. Key contributions include:
- Longitudinal Data Generation: The model generates realistic disease progression data by bridging gaps between sparse data points using a diffusion model, addressing challenges posed by irregular and infrequent subject visits.
- Conditioning on Time-Dependent Factors: It incorporates multiple time-dependent ordinal factors, such as age and disease severity, to generate data that reflects both cohort-level trends and individual-specific characteristics.
- Monotonic Longitudinal Modeling: Cohort-level characteristics are modeled using ordinal regression to capture monotonic progression over time.
- Experimental Validation: Extensive tests on four Alzheimer’s disease biomarkers demonstrate the model's superiority over nine baseline methods, underscoring its potential for broader applications in longitudinal data generation.

**Strengths:**

Clarity
- It is clearly presented.

**Weaknesses:**

The following are several concerns that can help further improve the methods and analyses.
- Most comparisons were conducted using the authors' proposed methods (Fig. 6), with only three simple baseline methods included in Fig. 7, which is insufficient. Numerous advanced longitudinal data analysis methods are available in the literature that could enhance the comparison.
- The reported F-measures are quite low. A straightforward approach, like using the current diagnosis to predict the next time point, might yield a better F-measure.
- The model assumes complete longitudinal data availability, which is a strong assumption and may not hold in real-world applications.
- The proposed model has numerous components, and a more detailed ablation study is needed.
- The manuscript contains numerous typos.

Originality
- The model was built based on empirical ideas instead of rigorous theory.

Quality
- The quality is okay but not elegant.

Significance
- The significance of the findings is low.

**Questions:**

Please help address the concerns shown above.

**Details Of Ethics Concerns:**

The data set with compete modalities tends to be small and skewed. There might be some disparity in the training and testing data.

---

### Official Review · Reviewer_czy3 · 2024-11-04

**Soundness:** 2
**Presentation:** 3
**Contribution:** 1
**Rating:** 3
**Confidence:** 4

**Summary:**

In this paper the authors presents a method called LEI (Longitudinal Ensemble Integration) for sequential classification using multimodal data. The methods develops on the recent EI (Ensemble Integration) approach to extend it to longitudinal multimodal datasets. This is done by introducing an LSTM model within the previously proposed EI framework. Categorical cross entropy loss is then used to make classification. This method is then used for dementia detection problem.

**Strengths:**

The paper tries to address a critical problem in biomedicine which can also be extended to other problems where several modalities are used to make inferences. Often these modalities carry information that is locally present hence leveraging information from several modalities to make decisions in a challenging task. The authors introduce LEI to extend EI framework to longitudinal datasets. This makes the motivation clear.
The proposed method is also able to identify temporal features that are important for the classification task.
The paper is well written and the results are discussed thoroughly.
Several configurations of LEI framework have been thoroughly discussed. These configuration/strategy allow for flexibility in terms of usage for the LEI framework.

**Weaknesses:**

Although the paper is well written and motivated I find the paper lacking with respect to the standards of the conference and the selected primary area.
1. Lacking novelty : I fail to understand the novelty of the paper. The paper seems to be heavily reliant on the Ensemble Integration method and makes several reference of the method throughout the paper. The only separating elements seems to be the use of LSTM. Though the authors have suggested several configuration in which LEI could be used in. Which by itself makes the novelty of the proposed approach limited. Maybe a more model-based approach starting from a graphical model could help with this. I would also encourage authors to discuss the underlying assumptions behind the longitudinal ensemble integration. This would allow the reader to understand why the usage of LSTM is non trivial and suitable for longitudinal multimodal datasets.
2. The interpretable aspect of the LEI approach still seems to be static and reliant on the EI framework itself. This limits the ability of the LEI framework to find temporal signature that might be more informative for classification. I would further encourage the  authors to address this in each of the configuration (time dependent BPs and time distributed BPs). As these configuration treats temporal dependencies differently. This could further enhance and showcase the contribution of the method in different settings.
3. Comparing LEI framework with other approaches could help showcase LEI efficacy which is currently not very clear.

**Questions:**

Other than the weakness, some of my questions are listed below.
1. Did the authors try other types of dynamical models for the LEI method? What was the rationale behind choosing LSTM ?
2. For a multi-modality setting it is quite common to have missing modalities at different time points [1][2]. Would the LEI framework be able to provide comparable results ?
3. How does the inter-relation between several modalities affect the performance of the LEI approach?
4. Did the authors try LEI framework on other multimodality problems [2][3]?

[1] Zhang, Chaohe, et al. "M3care: Learning with missing modalities in multimodal healthcare data." Proceedings of the 28th ACM SIGKDD Conference on Knowledge Discovery and Data Mining. 2022.

[2] Akbar, Md Navid, et al. "Advancing post-traumatic seizure classification and biomarker identification: Information decomposition based multimodal fusion and explainable machine learning with missing neuroimaging data." Computerized Medical Imaging and Graphics 115 (2024): 102386.

[3]Zhao, Jinming, Ruichen Li, and Qin Jin. "Missing modality imagination network for emotion recognition with uncertain missing modalities." Proceedings of the 59th Annual Meeting of the Association for Computational Linguistics and the 11th International Joint Conference on Natural Language Processing (Volume 1: Long Papers). 2021.

---

### Meta-Review · Area_Chair_YUsj · 2024-12-15

**Metareview:**

This paper presents a method, "Longitudinal Ensemble Integration (LEI)", for sequential classification using multimodal data. All of reviewers agree that modeling multimodal longitudinal data is important in biomedicine. The paper is well written. However, there are a few critical concerns raised by reviewers. LEI builds on the previous EI framework but follows an intuitive approach to generating and aggregating base predictors, resulting in limited novelty in its integration method. In addition, empirical comparison, ablation study, should be improved. There are a lot of good suggestions you can find in reviewers' comments. Therefore, the paper is not recommended for acceptance in its current form. I hope authors found the review comments informative and can improve their paper by addressing these carefully in future submissions.

**Additional Comments On Reviewer Discussion:**

There was no author response. During the discussion period, there was no change and all reviewers stood by their original decisions.

---

### Decision · Program_Chairs · 2025-01-22

Reject